# Alcohol consumption among university students in ASEAN countries: A systematic review and meta-analysis

Mayank Kejriwal 

Information Sciences Institute, University of Southern California, Los Angeles, CA, USA

## Overview Review

alcohol; ASEAN; systematic review; meta-analysis; university students

**Corresponding author:**
Mayank Kejriwal;
Email: keriwal@isi.edu

## Abstract

Alcohol consumption among university students poses significant public health challenges, especially in the Association of Southeast Asian Nations (ASEAN) region, where limited research exists. This review aims to synthesize evidence on sociodemographic factors associated with alcohol consumption among university students in ASEAN countries, assess the study quality and identify research gaps. A systematic search across nine databases was conducted in May 2024, using *Population, Intervention, Comparator, Outcome, Study Design and Timeframe* to define the inclusion criteria. Studies were assessed for quality and risk of bias using the AXIS tool. Data on sociodemographic factors were extracted, and random-effects meta-analyses were performed for frequently reported factors. Heterogeneity was measured using Cochran's Q-test and I-squared statistic, and small-study bias was tested using funnel plots and Egger's test. Fifteen cross-sectional studies involving 35,527 participants met the inclusion criteria. Gender, age and parental alcohol consumption were the most commonly studied factors. Male students had three times the odds of consuming alcohol compared to female students, a result robust to sensitivity analysis. Parental alcohol use and older age were also significantly and positively associated with alcohol consumption, with minimal heterogeneity. Most studies were of high quality, although variability in study design and geographic representation limited the generalizability of the findings. Sociodemographic factors such as gender, age and parental alcohol consumption influence alcohol use among ASEAN college students. However, cross-sectional design and limited country representation highlight the need for further robust research to inform policy and interventions.

## Impact statement

This review fills a critical gap in the understanding of alcohol consumption patterns among university students in the Association of Southeast Asian Nations (ASEAN), a population with unique sociocultural dynamics and limited prior investigations. By systematically analyzing sociodemographic factors such as gender, age and parental alcohol consumption, this study aims to provide actionable insights for policymakers, educators and public health practitioners seeking to design targeted interventions.

The findings reveal that male students are significantly more likely to consume alcohol than their female counterparts within ASEAN countries. Positive associations are found between consuming alcohol and both older age (within the university cohort) and parental consumption of alcohol. The review suggests the importance of culturally tailored, family-inclusive educational campaigns and policy measures aimed at reducing alcohol consumption among students. Furthermore, the identification of gaps in geographic representation and study designs offers a roadmap for future research to ensure more comprehensive regional coverage and robust methodologies.

The wider impact of this research lies in its potential to inform evidence-based policies and interventions that can improve student well-being and reduce the public health burden associated with alcohol misuse. At the regional level, it supports the United Nations' agenda on youth health and safety. Internationally, the study contributes to the global understanding of alcohol consumption behaviors in diverse cultural contexts.

## Introduction

Health-risk behaviors are actions that can lead to an increased risk of diseases and injuries (Surís et al., 2008; Peltzer and Pengpid, 2016), and vary across different age groups, environments and cultures (Duell et al., 2018; Rattay et al., 2018; Wattanapisit et al., 2020). Among these, alcohol consumption is a significant concern, particularly for university students (Mekonen et al., 2017). In developed and emerging economies alike, university students, especially first-year students

who are in early adulthood and transitioning from high school to university, have long been observed to engage in excessive alcohol use (Gill, 2002; Hebden et al., 2015). Habits formed during these years may also have long-lasting effects on health, potentially increasing the likelihood of noncommunicable diseases (NCDs) and chronic ailments in later life, such as diabetes, heart disease, stroke and certain cancers ( Shield et al., 2014; Budreviciute et al., 2020). The burden of NCDs in emerging economies has been increasing in recent decades (Boutayeb and Boutayeb, 2005; WHO, 2018) and reducing alcohol abuse is an explicit goal of the United Nation's Sustainable Development Goal (SDG) 3.5 (Flor and Gakidou, 2020).

More importantly, such behaviors are modifiable and preventable. While more intense forms of alcohol consumption, such as *heavy episodic* or *binge* drinking (defined by the World Health Organization as consuming at least 60 g of pure alcohol on a single occasion within the past 30 days), are significantly associated with alcohol-related harm (Moure-Rodríguez et al., 2014), researchers have found it easier to measure and study ordinary alcohol consumption (Yi et al., 2017; Wattanapisit et al., 2022), with which binge drinking is also correlated (Caffrey et al., 1996).

Focusing specifically on alcohol consumption among university students, Peltzer and Pengpid (2016) cite it as a significant public health issue, but also note that most research focuses on North American and European populations (Dantzer et al., 2006; Wicki et al., 2010; Perera and Torabi, 2012; Venegas et al., 2012). In developing economies, according to Peltzer and Pengpid (2016), estimates of drinking alcohol among university students range from 16.7% (China) to 48.8% (Malawi) for males, and from 3% (South Africa) to 24% (Colombia) for females. Other available evidence, while limited, also suggests that the prevalence of hazardous drinking in developing economies may be approaching that of industrialized countries. Taking an international perspective and reviewing published articles from 2005 to 2006 on alcohol use among university students Karam et al. (2007) states that "in Europe, Australasia and South America, college student drinking is problematic, to an extent similar to that reported in North America." More recently, Francis et al. (2014) conduct a systematic review of alcohol use among youth in East African countries, and find a 33% prevalence rate for university students, which was second only to male sex workers (69%).

Such observations highlight the need for more research into alcohol-related health risks across diverse global contexts. One such context is the set of rapidly developing and influential economies in the Association of Southeast Asian Nations (ASEAN), a regional intergovernmental organization founded in 1967 with shared goals and commitments to address challenges in areas such as human rights and public health (Lamy and Phua, 2012). Its founding members were Indonesia, Malaysia, the Philippines, Singapore and Thailand, and it has since expanded to include 10 member states, including (besides the ones above) Brunei Darussalam, Vietnam, Laos, Myanmar and Cambodia.

The population in ASEAN economies is young, educated and increasingly globalized, leading to a concurrent rise of modern health challenges, such as alcohol addiction among university students. In an influential study of 15,366 university students from seven ASEAN countries, Wattanapisit et al. (2022) identify "alcohol drinker" as one of five clusters of health-risk behaviors and find that the prevalence of alcohol use exceeds 10%. Similarly, Yi et al. (2017) survey 8,809 university students from nine ASEAN countries and determine a prevalence ranging between 12.8% (infrequent drinking) and 6.4% (binge drinking). At the country-specific level,

prevalence estimates can vary significantly. Wattanapisit et al. (2022) find that the odds of Singaporean university students consuming alcohol are 14 times that of students in Brunei, whereas, in Islamic countries like Indonesia and Malaysia, the odds are significantly lower.

Developing a better understanding of the sociodemographic drivers contributing to alcohol consumption among university students in ASEAN countries is a well-motivated and timely research agenda – one that could be critical for formulating effective and sustained strategies, policies and interventions. In other regions, several studies have shown positive associations between some sociodemographic determinants and alcohol consumption among youth. In the United Kingdom and Ireland, a systematic review found that gender was correlated with alcohol use disorder. However, the authors noted that the gender gap has narrowed recently (Davoren et al., 2016). Other sociodemographic determinants that have been studied include factors such as whether the student lives on campus, is a member of a sports club, has parents who drink, academic performance and age (Mekonen et al., 2017; Wattanapisit et al., 2020). High alcohol consumption among university students is now known to be associated with a range of negative outcomes, including injuries, violence, academic underperformance, long-term health complications and increased healthcare burdens (White and Hingson, 2014; Paul et al., 2024), highlighting the urgent need for early prevention and policy intervention among university youth.

This study aims to conduct a systematic review of the sociodemographic factors associated with alcohol consumption among university students in ASEAN countries. The specific objectives are to (a) determine specific sociodemographic factors that have been found to be positively or negatively associated with alcohol consumption in university students within ASEAN countries; (b) compare these studies in terms of quality, rigor and key findings and (c) conduct a meta-analysis of sociodemographic factors found to be most commonly studied.

## Methods

### Design

The *Population, Intervention, Comparator, Outcome, Study Design and Timeframe* framework was used to define inclusion criteria, search concepts and research questions, as well as to guide decisions on study eligibility, data extraction and analysis (Appendix C of the Supplementary Material).

### Inclusion and exclusion criteria

Following the PRISMA guidelines of (Moher et al., 2009) and the Cochrane Handbook of (Higgins et al., 2019), this review focuses on university students within ASEAN countries (Indonesia, Malaysia, the Philippines, Singapore, Thailand, Brunei, Vietnam, Laos, Myanmar and Cambodia). The primary outcome is *alcohol consumption*. In most of the included studies, alcohol consumption was treated as a binary outcome (*drinker* vs. *nondrinker*), often assessed through self-report or validated tools such as the Alcohol Use Disorders Identification Test (AUDIT). Variations in measurement were documented, and where necessary, outcomes were harmonized into a binary framework to enable meaningful comparison across studies.

Studies reporting sociodemographic exposures (e.g., gender, age, income, religion, peer influence) are prioritized, with students

who do not consume alcohol as the comparator group. Both observational and interventional study designs were considered, without publication date restrictions, although recent studies (1990–2024) were prioritized. Only English-language, peer-reviewed articles and relevant reports (e.g., from WHO) were included, acknowledging the limitation of excluding non-English papers from the region. Studies only involving other drugs or substance use addictions were specifically excluded, as were studies not involving university students as the primary (or, at minimum, separately studied) population of interest. Conference abstracts, gray literature, opinion articles, editorials and review articles, including systematic reviews and meta-analyses, were excluded from the analysis.

### Data

Nine databases were searched for published peer-reviewed articles with quantitative data, encompassing international (PubMed, Web of Science, Scopus, Medline, Embase, Cochrane Library), regional (Global Health, Global Index Medicus) and country-specific (Garuda Rujukan Digital, focused on Indonesia) sources to ensure broad coverage. The search strategy for each database is detailed in Appendix A of the Supplementary Material, along with a detailed count of documents found (where possible), thereby establishing a comparative baseline between the databases in terms of search coverage. Both thesaurus searching (e.g., using MeSH terms) and free-text searches were used. Although the search strategy mostly utilized international databases, only studies focused on ASEAN populations were included for the review and meta-analysis, as a key objective of the study is to fill ASEAN-specific knowledge gaps on alcohol use disorder among university students. Nevertheless, in the subsequent discussion, we do compare our findings with selected systematic reviews that have performed a similar analysis in other international contexts.

The references from the identified studies were also reviewed to identify other potentially relevant studies. The general strategy was designed to be inclusive, with no time limits applied to the search, and with synonyms, subject headings and Boolean operators like OR and AND judiciously used to obtain an initial set that could be reviewed for further inclusion. Specific inclusion and exclusion criteria are discussed next. The search was carried out between May 20 and 25, 2024. EndNote 21 was used as the reference manager.

The initial search results were first imported, and duplicates were removed based on a judicious combination of identical references found by matching the title and author, followed by a cursory manual review. Titles and abstracts were then screened, with studies excluded if they did not meet the inclusion and exclusion criteria. The full-text articles were then evaluated more closely in accordance with the established criteria. The reference lists of eligible studies were reviewed for any additional relevant articles, and suggested or similar articles found online were then evaluated for eligibility.

Each included full-text article was comprehensively reviewed, and the following study characteristics were extracted and tabulated both for individual study synthesis and statistical analysis and meta-analysis, using a data extraction form: *first author and publication year, sample size, percentage of sample size that is female, period of study, country, age range of participants in years (with mean and standard deviation, where available), study design* and *relevant details on sampling method.*

All included studies were ultimately found to be cross-sectional in design, and each study was further analyzed manually to extract the sociodemographic characteristics and exposures in the study as well as specific details on the outcome measured and key findings of interest distilled from the full text. These are tabulated separately, with a brief narrative synthesis provided. For sociodemographic characteristics that are studied in a sufficient number of studies, associations are further subjected to quantitative analysis and meta-analysis.

### Statistical analysis and heterogeneity

All statistical analysis was conducted using Stata 17, and meta-analysis on each identified sociodemographic characteristic was conducted for which a sufficient number of studies were available (typically, $n > = 5$, but exceptions could be made for $n = 4$, with the caveat noted in the results that it is a small-scale meta-analysis in the results, and not necessarily generalizable). Considering the outcome measure (alcohol consumption), the specific way in which it is measured could differ (e.g., the form of the question that is asked) across studies, which is a likely source of heterogeneity in the meta-analysis. Therefore, an important goal of the results is to provide sufficient details on the measurement of how alcohol consumption was measured.

Because the studies together span multiple countries and include substantial heterogeneity, the random-effects model was used for the meta-analysis. For completeness, a fixed-effects meta-analysis was also conducted and compared to the random-effects results. Heterogeneity is tested based on the methodology described by Higgins et al. (2019), with both the $I^2$ statistic and $p$-value from Cochran's Q test reported. Publication bias is measured using both funnel plots and Egger's test.

A systematic assessment of each study selected for the final review was conducted for quality and risk of bias, using the Appraisal tool for Cross-Sectional Studies (AXIS) instrument (Downes et al., 2016). A complete tabulation of the instrument for all included studies is reproduced in Appendix B of the Supplementary Material. The questionnaire includes 20 items spanning the major sections of the study (methods, results and so on). In using the tool for each cross-sectional study independently evaluated by it, a study is rated as *high quality* (low risk of bias), *medium quality* (moderate risk of bias) and *low quality* (high risk of bias) for total scores between 15–20, 10–14 and < 10, respectively.

### Results

A total of 10,351 studies were gathered from searching the nine selected databases, including an additional 23 records identified through citation-searching (Figure 1). Of these, 2,428 (23.46%) were found to be duplicates and were removed. Following this process, the remaining 7,923 records were screened for suitability based on titles and available abstracts. Of these, 163 (2.06%) publications were judged to be suitable for further analysis, but of these, the full-text of four (2.45%) publications could not be found or retrieved. After screening the remaining 159 publications for eligibility, a total of 15 (9.43%) studies were deemed eligible for final inclusion in this systematic review based on screening conducted using the inclusion/exclusion criteria.

### Study descriptions

The 15 studies included in this review encompassed a total sample of 35,527 individuals and collectively included all ASEAN countries, although some countries were more represented than others. All studies were cross-sectional in design and primarily used self-

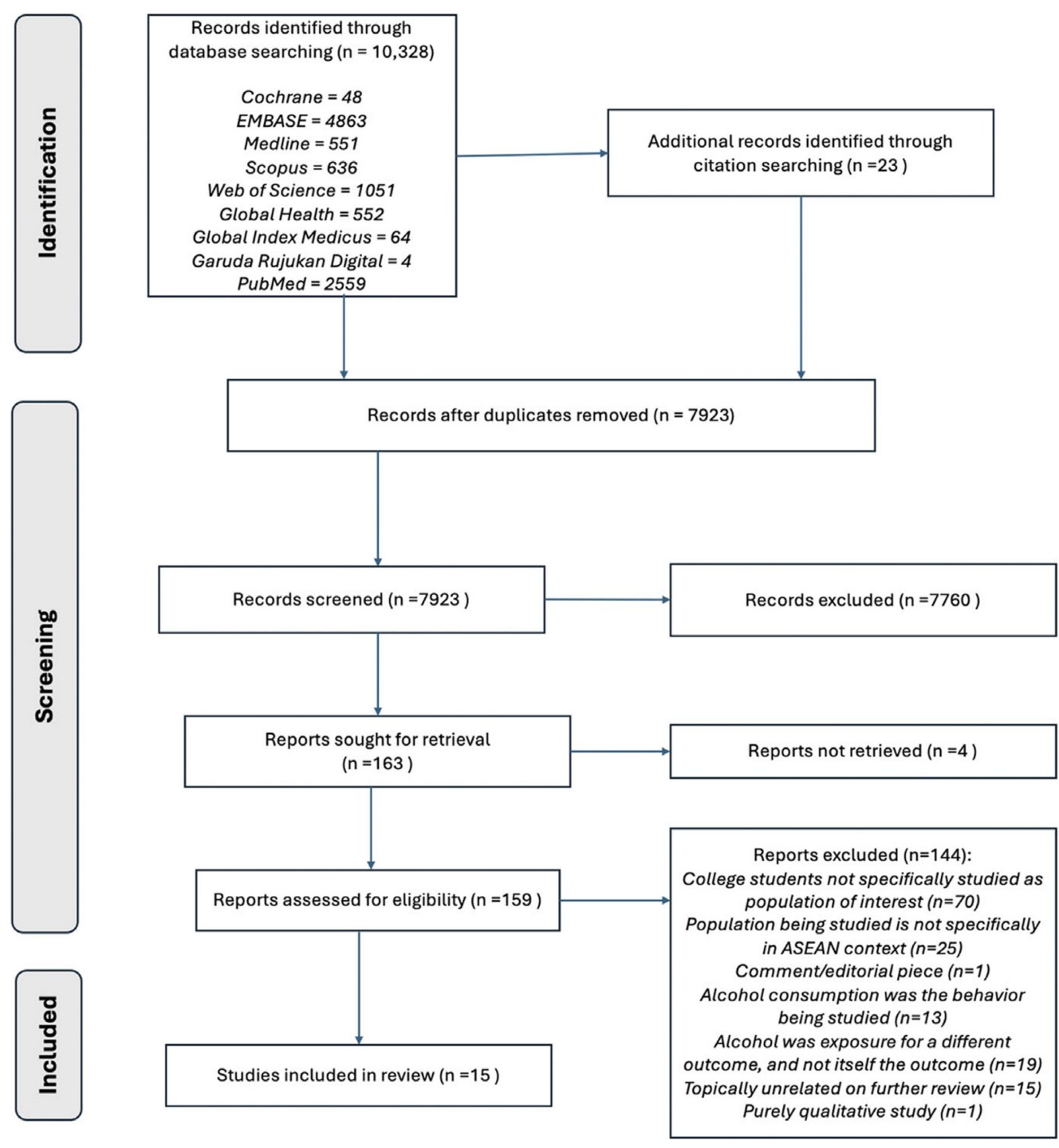

**Figure 1.** PRISMA flow diagram.

reported surveys as their instruments. The basic characteristics of the included studies are described in Table 1. Two studies spanned multiple countries, with Yi et al. (2017) covering nine ASEAN countries (Cambodia, Indonesia, Laos, Malaysia, Myanmar, the Philippines, Singapore, Thailand and Vietnam) and Wattanapisit et al. (2022) covering seven ASEAN countries (Philippines, Singapore, Thailand, Vietnam, Malaysia, Indonesia, Brunei Darussalam). Of the other 13 studies, three (exclusively) covered Myanmar, six covered Thailand, and one study each covered Vietnam, Cambodia, Malaysia and Indonesia. Based on the search criteria, no studies were

found matching the criteria that exclusively covered the Philippines, Singapore, or Brunei Darussalam, although there is some literature (Lim et al., 2007; Pagkatipunan, 2017; Lee et al., 2020; Rahman et al., 2023) on the prevalence of alcohol consumption among the university population (but without any data on, or associations with, any sociodemographic variables), or covering other youth populations, such as school-going adolescents (Hong and Peltzer, 2019; Pengpid and Peltzer, 2019). With the exception of four studies, all studies reported the study period. Only one included study reported results before 2000 (Caffrey et al., 1996), with all other studies having been

**Table 1.** Descriptive characteristics of all cross-sectional studies included in review

| Study | Country | Total number of participants/sample size (% female) | Age range in years, mean with standard deviation (s.d.), where available | Study period | Relevant details on sampling method described in publication |
|---|---|---|---|---|---|
| Htet et al. (2020) | Myanmar | 3,456 (62%) | 15–24 years old, (mean = 18.7, s.d. = 1.6) | August 2018 | Two-stage probability sampling design with university students recruited from six universities in Mandalay, Myanmar. |
| Jaichuen et al. (2018) | Thailand | 1,279 (70.6%) | 17–25 years old | December 2013–March 2014 | Multistage stratified cluster sampling survey, with a formula given by Krejcie and Morgan (1970) used to determine the sample size of 1,279 university students from "all regions of Thailand." |
| Mathialagan and Teng (2017) | Malaysia | 150 (52%) | 18–31 years old | Not specified | Described as "purposive sampling" where participants were approached in SEGi College Subang Jaya. Malay students were purposely not approached as drinking was against their Islamic law. |
| Buakate et al. (2022) | Thailand | 685 (74.9%) | 18–22 years old, majority (81.61%) were aged between 18 and 20 | April 2019–September 2019 | The authors state that "685 of 1,100 first-year students from 12 universities in southern Thailand were randomized and recruited using eligible criteria." A rigorous methodology, using the formula by Krejcie and Morgan (1970), was used to compute sample size. |
| San San et al. (2010) | Myanmar | 400 (54%) | 15–24 years old (mean = 20 years) | June–July 2008 | Medical students who attended class during the date and time of data collection were specifically selected. At the Medical University, Yangon, 3,010 medical students (1,488 males and 1,522 females) were enrolled at data collection time. The sample size was 400 from each group was calculated using a "sample size calculation formula of different proportions." |
| Phoosuwan (2019) | Thailand | 853 (56%) | Majority were 18–29 years old (mean = 20.83 years old, s.d. = 1.45) | Not specified | Participants were undergraduate students recruited from universities in a Northeast province using stratified random sampling. A sample size formula was used for obtaining the 853 sample size. |
| Supit et al. (2017) | Indonesia | 417 (81%) | Not specified | June–July 2016 | A "total sampling" method was employed to ensure maximum representation from the population of all enrolled undergraduate college students in the School of Public Health, Manado State University, at the time of data collection. |
| Aung et al. (2019) | Myanmar | 976 (45.3%) | 16–19 years old, (mean = 18.46, s.d. = 0.05) | Not specified | Two universities from three districts in Mandalay region were randomly selected, from which 976 students (stratified by academic year and sex) responded to questionnaire. |
| Sok et al. (2020) | Cambodia | 1,359 (49.2%) | mean = 21.3 years, s.d. = 2.3 | June–July 2015 | Undergraduate students were randomly selected from the Royal University of Phnom Penh and University of Battambang. These universities were "purposively" selected to partake in this research. |
| Nguyen et al., (2019) | Vietnam | 263 (75%) | 18–24 years old (mean = 20.24, s.d. = 1.429) | 2022 | Convenience sampling from "a private university in a large urban city in Vietnam." |

(Continued)

**Table 1.** (*Continued*)

| Study | Country | Total number of participants/sample size (% female) | Age range in years, mean with standard deviation (s.d.), where available | Study period | Relevant details on sampling method described in publication |
|---|---|---|---|---|---|
| Caffrey et al. (1996) | Thailand | 501 (67%) | Not specified | 1993–1994 | A stratified cluster sampling strategy was used to originally sample approximately 10% of the student population. The final actual percentage sampled was 12.9. |
| Tonkuriman et al. (2019) | Thailand | 413 (45%) | 18–22 years old (mean = 20, s.d. = 1.185) | 2014 academic year | After receiving permission from the presidents of seven universities, the researcher recruited students, but only participants with binge drinking experience answered additional online questionnaire. |
| Boonchuaythanasit et al. (2021) | Thailand | 600 (not specified) | Mean = 20.83 years, s.d. = 1.5 | Not specified | Participants included representative first to fifth year undergraduate students at the Thailand National Sports University, with 600 students being selected using stratified random sampling procedures. |
| Yi et al. (2017) | 9 ASEAN countries: Cambodia, Indonesia, Laos, Malaysia, Myanmar, the Philippines, Singapore, Thailand and Vietnam | 8,809 (62%) | Mean = 20.5 years, s.d. = 2 | 2015 | A stratified random sampling procedure was used. One department was randomly selected from each faculty as a primary sampling unit, and from each selected department, students were randomly selected from all the courses. |
| Wattanapisit et al. (2022) | Seven ASEAN countries: Philippines, Singapore, Thailand, Vietnam, Malaysia, Indonesia, Brunei Darussalam | 15,366 (53%) | 18–22 years old, but majority were 19–21 years old | 2020–2021 | Sampling method was not explicitly specified, but the authors seem to suggest that the survey was sent to all undergraduate students in the 17 ASEAN-HPN member universities. Response rate is not specified. |

conducted over the last two decades (2008–2022). As expected, the age range of participants skews younger (typically between 18 and 24 years old), and in most studies, skews positively toward females.

### Identified sociodemographic characteristics, key associations and meta-analysis

In the context of this review's first objective, the data compiled in Table 2 shows that a wide variety of sociodemographic characteristics have been explored across the 15 included studies. Gender was the most commonly reported exposure, and most studies consistently found that being male was significantly associated with alcohol consumption compared to being female. In a subsequent meta-analysis, a pooled odds ratio is used to quantify the effect on alcohol consumption across studies. In the studies where binge drinking was reported or explicitly differentiated from alcohol consumption per se, the association still remained strong.

Other common sociodemographic variables explored in the studies, albeit less commonly so than gender, were age, Grade Point Average (GPA) or academic performance, living situation (e.g., whether participants lived with parents or in an on-campus dormitory), smoking and tobacco use, attitude toward alcohol consumption and parental alcohol consumption. San San et al. (2010) and Yi et al. (2017) highlighted that age and life satisfaction, combined with factors like peer influence and physical activity, affect the likelihood of alcohol consumption. Parental influence,

particularly the drinking behavior of parents, was repeatedly cited as a strong predictor of alcohol uses (San San et al., 2010; Phoosuwan, 2019). Less certain is the role of religion, academic performance and living situation, which showed variable influence, with religiosity not always significantly associated with alcohol consumption (Nguyen et al., 2024).

### Meta-analysis of recurring characteristics

The results of the random-effects meta-analysis for gender are shown in Figure 2a. An overall OR of 3.15 (95% CI: 2.24–4.06) was observed, which suggests that being male is significantly and strongly associated with alcohol consumption. However, the heterogeneity was substantial and significant ( $I^2$=85.2%, $p < 0.001$), suggesting that the results need to be treated with caution. This high heterogeneity is consistent with two other meta-analyses that have also measured associations between gender and alcohol consumption, including a meta-analysis by Nascimento et al. (2022) (who observed an $I^2$ statistic of over 95%) and Francis et al. (2014) (who also noted significant heterogeneity in all subgroups using the $I^2$ statistic, but unlike Nascimento et al. (2022) and this study, deferred from reporting a pooled association measure).

A sensitivity test was also conducted by excluding the studies by Sok et al. (2020) and Supit et al. (2017) and redoing the meta-analysis, as they seem to be outliers compared to the other studies. Although not illustrated herein, the heterogeneity was found to reduce to moderate levels when only the other six studies were considered ( $I^2$=57.6%, $p = 0.038$). The effect of being male

**Table 2.** Sociodemographic characteristics identified in each selected study, measurement of outcome and key results/associations distilled per study

| Study | Sociodemographic characteristics[a] | Measurement of outcome[a] | Key results and associations |
|---|---|---|---|
| Htet et al. (2020) | Sex[d], age, monthly expenses, parents' or guardians' alcohol consumption, peers' alcohol consumption, truancy, smoking in the previous 30 days, feeling sadness or hopeless in the previous 12 months | Alcohol consumption (binary), defined as "consuming at least one drink of alcohol (includes drinking a bottle of beer, a glass of wine, a glass of liquor such as whisky, rum, cocktail or mixed drink) on at least 1 day in the 30 days prior to the survey, but excludes drinking a few sips of wine for religious purposes." Responses were re-coded by authors on binary scale | (1) Alcohol consumption in the past 30 days was 20.3% overall (males: 36.0%, females: 10.8%) (2) Males had significantly higher odds of alcohol consumption (AOR[b] = 2.3, 95% CI[b]; 1.9–2.9) (3) Increased alcohol use was associated with truancy (AOR = 2.1, 95% CI; 1.3–3.3), smoking (AOR = 7.0, 95% CI; 5.1–9.7), feelings of hopelessness or sadness (AOR = 1.4, 95% CI; 1.2–1.8) and peers' alcohol use (AOR = 7.5, 95% CI; 4.8–11.7) |
| Jaichuen et al. (2018) | Sex, age, GPA score, province, preferences for sport, participation in sport, number of times exposed to alcohol advertising, positive attitude about alcohol | Students were asked whether they had ever consumed alcohol. Responses were "never consumed," "consumed but not in the past 12 months" and "consumed in the past 12 months." The latter two were classified as *current drinker* and *binge drinker*, respectively | Students participating in sport were more likely to drink alcohol (AOR = 1.7; 95% CI: 1.24–2.28) and binge drink (AOR = 1.6; 95% CI: 1.10–2.25) than students not participating in sport |
| Mathialagan and Teng (2017) | Parenting style | AUDIT[b] score: AUDIT consists of 10 items. The first 8 items were scored on a five-point scale and the last two items were three-point scale | Significant relation, using Pearson product–moment correlation coefficient, between authoritarian ($r = 0.246$, $p < 0.05$) and permissive ($r = -0.426$, $p < 0.05$) parenting style and alcohol consumption among the participants |
| Buakate et al. (2022) | Sex, age, religion, monthly income in baht, spending, living place, smoking, gambling, congenital disease, knowledge level, attitude level, marketing, self-competency, relative drinker | Alcohol consumption was tested through two items in questionnaire, combined into a single binary variable for logistic regression analysis | (1) Among other factors, Buddhism-practicing students were more likely to consume alcohol compared to Islam (AOR: 8.65, 95% CI: 4.41–16.97) (2) Smokers (including those who had quit) were also more likely, compared to nonsmokers (AOR: 17.9, 95% CI: 2.72–117.78) |
| San San et al. (2010) | Parents' drinking, close friends who drink, peer pressure to drink, experienced smoking tobacco, pocket money/day | Alcohol consumption was coded in the final analysis as a binary variable. The specific question and calculation are not provided in the paper | Age- and sex-adjusted ORs showed that parents' drinking, close friends' drinking, peer pressure to drink and smoking tobacco were all significantly associated with alcohol consumption |
| Phoosuwan (2019) | Gender, year of study, GPA, place of residence, father's occupation, mother's occupation, father's alcohol consumption, mother's alcohol consumption, sibling's alcohol consumption, attitude toward alcohol consumption | AUDIT (Thai version) score was computed and in the final analysis, coded as binary (drinker vs. nondrinker) | (1) Crude ORs showed that all the factors listed in Column 2 were significantly associated with alcohol consumption (2) Largest effect sizes were observed for mother's consumption of alcohol and attitude toward alcohol consumption |
| Supit et al. (2017) | Sex, year, current GPA, cumulative GPA, ethnicity, academic performance self-reflection importance of GPA | Alcohol consumption (binary) | (1) Using Kendall's tau, cumulative GPA was most strongly associated ($r = -0.211$, $p = 0.003$) with alcohol consumption (2) In crude OR analysis, academic performance self-reflection, importance of GPA and ethnicity were additionally found to be associated with outcome |
| Aung et al. (2019) | Gender, religion, education status, school activities, attendances, living situation, life satisfaction, smoking, betel chewing and parents' history | AUDIT score: a distribution was computed across four categories of scores[f], but for logistic regression, categories were condensed into binary (drinker vs. nondrinker) | Gender, smoking habit and living situation for drinking were significant predictors of alcohol consumption among university students |
| Sok et al. (2020) | Gender | Alcohol consumption (three-level categorical): "Would you describe yourself as a nondrinker, occasional drinker or regular drinker?" | Compared to nondrinkers, males were significantly more likely than females both to be occasional drinkers (OR = 8.02, 95% CI: 5.99–10.74) and regular drinkers (OR = 8.51, 95% CI: 5.57–12.99) |
| T Nguyen et al. (2019) | Religiosity[c], age, sex, peer substance use | Alcohol consumption encoded as binary variable: participants were asked if they had ever used beer, alcohol, wine and other alcoholic beverages | After controlling for age and sex, religiosity was not found to be significantly associated with consumption, but peer substance use was (AOR = 2.47, 95% CI: 1.87–3.26) |

(Continued)

**Table 2.** (*Continued*)

| Study | Sociodemographic characteristics[a] | Measurement of outcome[a] | Key results and associations |
|---|---|---|---|
| Caffrey et al. (1996) | Gender | A US-based research questionnaire, based on the Core Alcohol and Drug Survey, was translated into Thai with some modifications to questions. The raw responses were then used to encode for (binary) alcohol consumption for OR analysis | (1) Prevalence of alcohol consumption was higher in males than in females (2) The number of students reporting family members with alcohol-related issues was notably significant; however, Thai students were found to consume drugs or alcohol less frequently than their U.S. counterparts |
| Tonkuriman et al. (2019) | Attitude toward binge drinking, alcohol expectancy, binge-drinking refusal self-efficacy, peer influence, alcohol advertising and physical environment | A binge drinking behavior questionnaire was used to ask about the alcoholic drinks students consumed at least once in the past 30 days to calculate their standard drinks and to identify heavy drinking | Structural equational (causal) modeling showed that "binge drinking refusal self-efficacy" ($\beta = -0.22$, $p < .001$) and "peer influence" ($\beta = -0.14$, $p < .05$) were significant negative factors and "physical environments" ($\beta = 0.18$, $p < .001$) was a positive predictor of binge drinking |
| Boonchuaythanasit et al. (2021) | Health literacy (HL), drinking refusal self-efficacy (DRSE), alcohol expectancy (AE) | Alcohol consumption behavior (ACB) score, derived using AUDIT | Using coefficients of determination in a structural equation model, the Column 2 factors together accounted for 80% of the variance in the student's alcohol consumption behavior |
| Yi et al. (2017) | Age group, living situation, family economic status, country category, level of involvement in organized religious activity, level of involvement in non-organized religious activity, health knowledge and beliefs, level of life satisfaction, current tobacco use, past year illicit drug use, depressive symptoms and physical activity | Alcohol consumption[e] measured using the question: "How often do you have (for men) five or more and (for women) four or more drinks on one occasion?" Response options were 0 = never, 1 = less than monthly, 2 = monthly, 3 = weekly and 4 = daily or almost daily | After adjustment, among both males and females, higher binge drinking was independently and significantly associated with being in older age groups, lower level of non-organized religious activity, lack of knowledge on alcohol-heart disease relationship, weak beliefs in the importance of limiting alcohol use, lower level of life satisfaction, tobacco and illicit drug use, depressive symptoms and high level physical activity |
| Wattanapisit et al. (2022) | Gender, age, year of study, BMI, country, GPA, place of living, commute time, commute type, housing type and member of sports club | Current drinker (binary), based on a single question directly asking whether the participant currently drinks | Students in the Philippines, Singapore, Thailand and Vietnam were more likely to consume alcohol, compared with those in Brunei |

[a]Unless otherwise specified through a footnote, all sociodemographic characteristics and measurement of outcome were obtained through self-administered/self-reported questionnaire.
[b]Abbreviations: AUDIT: Alcohol Use Disorder Identification Test (Babor et al., 2001), AOR: Adjusted Odds Ratio, CI: Confidence Interval, OR: Odds Ratio.
[c]The authors defined religiosity as a "latent construct" examining "how religious the participants are." It is computed as a sum of mean scores of five specific items on the questionnaire.
[d]The same terminology (sex or gender) is used here as specified in the cited study.
[e]Although the authors purported to measure binge drinking, in logistic regressions, they combined *infrequent* and *frequent* binge drinking into a single category that has been referred to here simply as *alcohol consumption*.
[f]AUDIT four-category model: Abstainers (0 score), low risk drinkers (1–7 scores), high risk drinkers (8–15 scores) and dependence (16–40 scores).

remained strong and significant, but was reduced (overall OR = 2.28, 95% CI: 1.78–2.77). These ORs are generally consistent with the association measures found in the literature for other regions: Nascimento et al. (2022) find that, among Brazilian medical students, 65% of male students engage in binge drinking compared to 47% for women. Results compiled by Peltzer and Pengpid (2016) find similar gender-specific prevalence differences in Colombia (46% male vs. 24% female), China (ranging from 16.7 to 37.4% male vs. 5.4 to 11.6% female), Malawi (48.8% male vs. 5.0% female), South Africa (27% male vs. 3% female), Thailand (32% male vs. 7% female), Uganda (34.1% male vs. 23.4% female) and Venezuela (32% male vs. 15% female). In almost all of these cases, the prevalence of alcohol consumption among males is at least two to three times higher than among females.

In a second sensitivity analysis, where the multicountry surveys by Yi et al. (2017) and Wattanapisit et al. (2022) were included, heterogeneity went up substantially ($I^2$=93.4%, $p < 0.001$), but overall OR remained at commensurate levels compared to earlier analyses (OR = 2.49, 95% CI: 1.88–3.10). Together, these analyses confirm that the odds of alcohol consumption by college students in

ASEAN countries remain two to three times higher for male students, compared to females.

Figure 2b uses the random-effects model to show that the overall odds of consuming alcohol are 1.5 times (overall OR = 1.5, 95% CI: 1.28–1.72) for students in the oldest age group in college (who still tend to be young; typically in early-mid 20s) compared to the youngest. However, there are only four studies that could be included, so the results should be cautiously interpreted as arising from a small-scale meta-analysis. Heterogeneity is nonexistent ($I^2$=0%, $p = 0.554$), in stark contrast with the gender analysis. Fixed-effects meta-analysis, while not reproduced in the figure, showed that the overall OR, 95% confidence intervals and heterogeneity were unchanged.

The only other sociodemographic variable for which enough studies were available for conducting meta-analysis was parental consumption of alcohol. Random-effects results, demonstrated in Figure 2c, show that the overall odds of consuming alcohol were 1.58 times higher (OR = 1.58, 95% CI: 1.31–1.85) for those with parents consuming alcohol, compared to those whose parents did not consume alcohol. Heterogeneity was again low ($I^2$=13.6%,

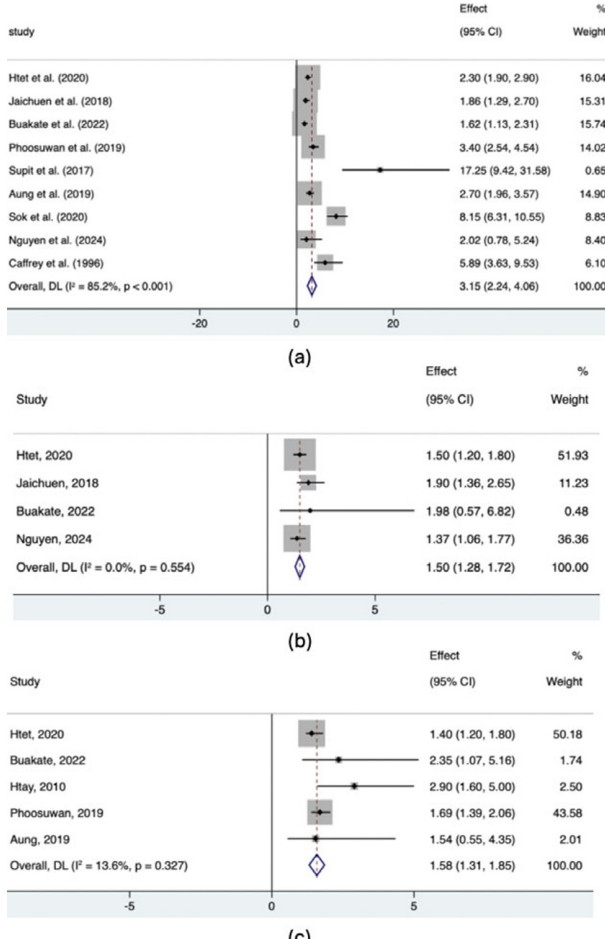

**Figure 2.** Association between consuming alcohol and (a) gender (male vs. female), (b) age (oldest age group vs. youngest age group) and (c) parental consumption of alcohol (having parents who consume alcohol vs. parents not consuming alcohol) using random-effects pooling of odds ratios (ORs) identified in relevant selected studies.

$p = 0.327$) and not significant. Robustness analysis using the fixed-effects model led to similar conclusions, with heterogeneity unchanged but the overall OR experiencing a slight decline (OR = 1.56, 95% CI: 1.34–1.78).

### Quality assessment

Using the AXIS instrument (Appendix B of the Supplementary Material), eight studies were classified as high quality, with scores ranging from 17 to 19, indicating robust study design, appropriate population sampling and well-reported methodologies. Notably, the study by Aung et al. (2019) achieved the highest score of 19, with strengths in sample representativeness and efforts to address non-responders. Other high-quality studies, such as those by Buakate et al. (2022), Htet et al. (2020) and Jaichuen et al. (2018) displayed strong adherence to AXIS guidelines, including clear objectives and reliable outcome measures. However, certain areas for improvement, such as better reporting of nonresponse rates and justifications for sample sizes, were noted even in these higher scoring studies. A common issue across almost all studies was the inadequate addressing potential response bias, and the lack of attempting to characterize nonresponse. A smaller number of studies were categorized as medium quality, scoring between 10 and 14.

Weakness in these studies included a lack of justification for their sample size or an inadequate discussion of the study's limitations (Mathialagan and Teng, 2017; Supit et al., 2017; Nguyen et al., 2024). Overall, however, the studies' internal consistency and appropriate use of statistical methods still support their inclusion in the review. Both Tonkuriman et al. (2019) and Boonchuaythanasit et al. (2021) use a structural equation model, with associations that cannot be reconciled with the odds ratios prevalent in other models; hence, they are excluded from meta-analyses. Studies by Yi et al. (2017) and Wattanapisit et al. (2022) are both multicountry, in contrast with all other studies, and do not provide separate associations per country. They are also excluded from initial meta-analyses but are included in appropriate sensitivity analyses.

### Publication bias

For evaluating the risk of bias across selected studies, funnel plots were generated for each of the three common sociodemographic characteristics studied earlier in the meta-analysis (gender, age and parental alcohol consumption). As illustrated in Figure 3, for two of the sociodemographic characteristics (age and parental alcohol consumption), studies tended to be located within the triangle region and scattered close to the line of $log(OR) = 1$. The situation is slightly more complex for gender: several studies are located outside the triangle region; however, despite this scattering, there is symmetry on either side. Using Egger's regression test for small-study effects to formally evaluate publication bias, we obtained $p = 0.764$ (gender), $p = 0.447$ (age) and $p = 0.171$ (parental alcohol consumption). Because all three $p$-values are greater than 0.05, there is little evidence against the null hypothesis of symmetry (indicating no publication bias), and hence, publication bias in the previously reported meta-analyses is not present.

### Discussion

Our key objectives were to determine both the set and strengths of sociodemographic variables found to be associated with alcohol consumption among university students in ASEAN countries. In performing a literature search for the systematic review, we found 15 eligible cross-sectional studies covering all the ASEAN countries, either individually or as part of a comprehensive study sampling students across countries (Figure 1). Each study was varied in its coverage and measurements of sociodemographic characteristics (Table 1). Age, gender and parental alcohol consumption were among the most frequently occurring characteristics identified, although characteristics like religiosity and academic performance were also observed in more than one study. In distilling other key findings from these studies (Table 2), peer influence and smoking behavior were also found to be recurrent factors that significantly predicted alcohol use among students (Htet et al., 2020; Buakate et al., 2022).

While no other systematic review and meta-analysis of sociodemographic factors exists, to our knowledge, of alcohol consumption among university students in ASEAN countries, a preliminary survey of related reviews (e.g., in other countries, or covering youth populations other than university students) suggests that some of the main findings distilled from this review are consistent with those by Francis et al. (2014), Karam et al. (2007), Martens et al. (2006) and Wicki et al. (2010). For example, Karam et al. (2007) take a comparative international perspective, and reviews, relative to North America, "similar risk factors and protective factors" (including male gender and family use of alcohol) in regions as

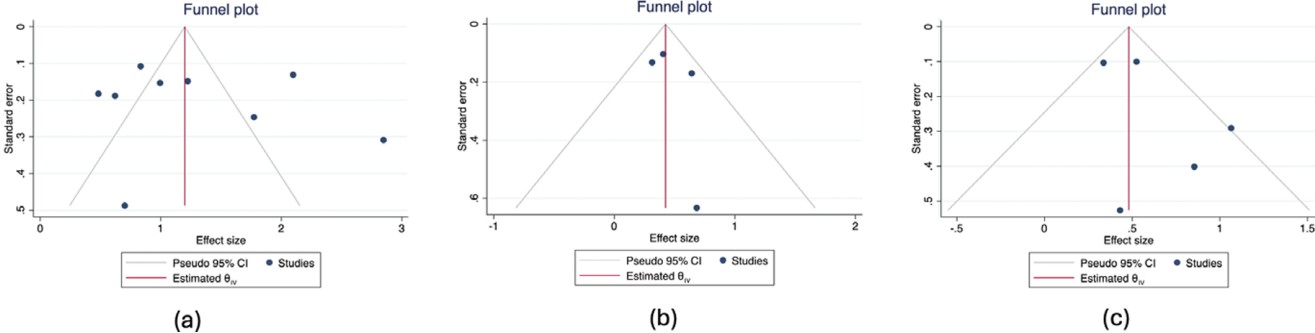

**Figure 3.** Funnel plot for assessing publication bias in studies included in the meta-analysis (Figure 1) for (a) gender, (b) age and (c) parental alcohol consumption.

diverse as Australasia, North and South America, Africa and Asia. Although their study did not include a meta-analysis, it confirmed significant associations between alcohol use and sociodemographic factors, including male gender, higher socioeconomic status, greater family educational attainment and excessive alcohol use by family members or peers (Karam et al., 2007).

Another related review by Francis et al. (2014) conducted both a systematic review and meta-analysis of alcohol use prevalence among youth in eastern Africa. They also found that most studies were cross-sectional in design, and because they considered a broader population group (including school students, as well as university students, street children and sex workers), were able to determine that the prevalence of alcohol use was highest among university students and male sex workers compared to other groups. Specifically considering gender-specific differences, they find that, while male college students had higher odds of consuming alcohol, female alcohol prevalence was higher both in primary school and in street children. These findings caution against whole-sale statements about gender differences in alcohol consumption. In yet another systematic review conducted for medical university students in Brazil, Nascimento et al. (2022) also find gender-specific differences, although they do not study other sociodemographic correlates of alcohol use. Gender-specific differences are also noted in reviews conducted for university populations in industrialized economies like the UK and Ireland (Martens et al., 2006; Davoren et al., 2016), but Davoren et al. (2016) argue in their review that the gap may be narrowing.

Our study has some limitations that need to be borne in mind. A consistent limitation is the studies' cross-sectional design, which only provides a snapshot of behaviors and associated factors at a single point in time, making it impossible to determine whether a factor, such as peer influence or academic performance, precedes or follows alcohol consumption. Hence, there is a need for longitudinal studies to better understand the temporal relationships and potential causal pathways between these factors and alcohol use among college students in ASEAN countries. Even for observational studies, a diverse range of designs (including cohort and case–control) may help provide deeper insights into alcohol use disorder.

Another limitation is the studies' general reliance on self-reported measures of alcohol consumption and related behaviors, which can introduce biases such as social desirability, recall error, or underreporting. Selection bias may also occur if students consuming alcohol are missing class, and hence had less probability of being selected for the survey. More minor concerns include heterogeneity in how questions on the survey are designed, and the lack of distinction or nuance in most studies between binge drinking,

alcohol addiction and alcohol consumption. This review had to rely on the last of these categories, as it was most commonly reported, but future research could better aim to distinguish between the three. Last but not least, the generalizability of the findings should be treated with caution, because several studies included in the review used convenience samples from specific universities or cities, limiting their broader applicability.

The findings of this systematic review have implications for both clinicians and policymakers. Early identification of high-risk groups can enable the development of more tailored, preventive health measures, particularly in student health services. Considering the positive association between being a male university student and consuming alcohol, university decision-makers need to recognize the potential consequences of alcoholism among male students, such as an elevated risk of injuries, violence and academic underperformance Paul et al. (2024). As prevention policies, universities could implement routine alcohol use screening for males during health checkups, incorporate brief alcohol intervention programs into campus health centers and train counselors to assess peer influence and smoking status as risk indicators, especially for males deemed to be at higher risk. In the literature, there is some precedent for each of these interventions in other contexts (Barnett and Read, 2005; Samson and Tanner-Smith, 2015). Similarly, given the positive association between older age (within the university cohort) and alcohol consumption, educational campaigns geared toward younger or incoming students on the risks of alcohol consumption and resources specifically targeted toward substance abuse disorders and addictions, may yield dividends for reducing alcoholism (Grossbard et al., 2016).

For policymakers, the review suggests the need for region-specific public health initiatives that address the broader social influences on drinking behavior, such as peer pressure. Policy-makers may consider implementing stricter regulations on alcohol advertising, mandating campus-based substance use programs and engaging parents and communities in prevention efforts. To be truly effective, such implementations would ideally be *culturally tailored* (Manuel et al., 2015). An example of one such intervention that was trialed in three Native American communities by McDonell et al. (2016) is culturally adapted contingency management, defined as an "addiction intervention where participants receive reinforcers such as vouchers or prizes for providing objective evidence of drug abstinence" and was cited as an effective intervention for illicit drugs, compared to other psychosocial interventions (Lussier et al., 2006; Prendergast et al., 2006). However, it needed to be adapted to the specific cultural setting of the Native American reservation to be feasible. Another example of a culturally tailored intervention is peer-to-peer-based motivational interviewing for alcoholism in the

context of American Indian Alaska Native women of childbearing age (Montag et al., 2017), the design process of which "included various community focus groups, interviews and a final review." While a number of such adaptations have been proposed both for Native Americans (Richer and Roddy, 2022), and also for specific demographic groups like Latino males (Valdez et al., 2018), little work exists on culturally tailored interventions for tackling alcohol use disorders in the ASEAN countries, despite their promise (Jiang et al., 2018).

An important aspect of this study and the formulated research objectives is that it treated alcohol consumption as an outcome only, thereby excluding the many studies where alcohol consumption was an exposure for another outcome e.g., hypertension or depression (Peltzer et al., 2017; Tuyen et al., 2019; Vo et al., 2023; Yeo et al., 2024). While this was by design, both because it is necessary to scope the study and because it is generally difficult to isolate the association between alcohol and the outcome variable in the presence of other controls (if alcohol is itself an exposure), real-life policy decisions would need to consider a more holistic picture when designing interventions. This study, while not making causal inferences, suggests some of the risk factors that could be taken into account by policymakers and those in authority in universities (such as Deans) to reduce the harmful effects of alcohol consumption among their student body.

## Conclusion

This systematic review aimed to provide a comprehensive analysis of sociodemographic factors associated with alcohol consumption among college students in ASEAN countries. Through the inclusion of 15 cross-sectional studies, key variables such as gender, age, parental alcohol consumption, peer influence and smoking behavior emerged as significant correlates of alcohol use. The review also identified gaps in the literature, including underexplored sociodemographic variables. As countries across the Global South experience rapid growth in their university populations, region-specific insights into alcohol consumption are essential for designing effective public health policies that reduce future healthcare burdens. By highlighting ASEAN-specific risk factors, the review's findings contribute to the global understanding of youth alcohol consumption and can inform both local and international intervention strategies, aligning with the United Nation's SDG 3 on reducing harmful substance use worldwide. Future research should further integrate regional specificity with global perspectives, promoting cross-national comparisons that strengthen evidence-based public health policy.

**Open peer review.** To view the open peer review materials for this article, please visit http://doi.org/10.1017/gmh.2025.10027.

**Supplementary material.** The supplementary material for this article can be found at http://doi.org/10.1017/gmh.2025.10027.

**Data availability statement.** Data availability is not applicable to this article as no new data were created or analyzed in this study.

**Author contribution.** M.K. is responsible for all aspects of this study.

**Financial support.** This research received no specific grant from any funding agency, commercial or not-for-profit sectors.

**Competing interests.** The authors declare none.

**Ethics statement.** As this is a systematic review, it did not involve any direct data collection from human subjects.

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
