## [Reviewer Report]

Overall Comments:

This study examining correlates of alcohol consumption among ASEAN university students is valuable, particularly given the authors' claim of a lack of prior reviews in this region. While the manuscript is generally well-written, several points require clarification and expansion to enhance its impact.

Specific Comments:

Impact Statement:

• Please specify the direction of the association between age, parental alcohol use, and alcohol consumption among university students (i.e., positive or negative correlation).

• Expand on the concept of ‘culturally-tailored campaigns’ by providing concrete examples. If this section has a limited space, expand somewhere esle. Specifically, clarify which of the identified factors (gender, age, parental alcohol consumption) inform this recommendation."

Introduction:

• Strengthen the rationale by including data on the prevalence of alcohol consumption, harmful use, or binge drinking among university students in the ASEAN region. Highlight the current impacts and problems associated with this issue to emphasize the study’s necessity.

• Reduce the emphasis on NCDs; a brief mention is sufficient.

• Update older references (e.g., 2002) with more recent sources.

Methods:

• Clearly state the inclusion criteria for publication years and the timeframe for data retrieval.

• Define the outcome variable precisely. Is it simply ‘drinker’ vs. ‘non-drinker,’ or are other measurements used? Address how the study handled variations in outcome variable measurement across included papers, especially in relation to comparison and meta-analysis.

Discussion:

• Enhance the discussion by comparing and contrasting the study’s findings with research from other regions.

• Revise recommendations in lines 466-476 for clarity and practicality. The suggestion of ‘targeted screening’ requires specific actionable steps for university decision-makers. Expand on how universities can implement prevention policies among male students. Additionally, address the lack of discussion regarding the potential consequences of higher alcohol consumption in males, such as injuries and disabilities.

• Although the results are based on cross-sectional study design, reviewer encourages the authors to explore possibly actionable recommendations for the other identified factors (age and parental alcohol use). Suggest potential interventions for health professionals or universities to address these factors.

Conclusion:

• Shorten the conclusion by focusing on the key findings. Avoid introducing new information or expanding on discussion points. Maintain conciseness and clarity.

---

## [Reviewer Report]

The manuscript is well-written and presents the socio-demographic factors associated with alcohol consumption among college students in ASEAN countries.

Following are the minor comments on the paper from my side:

1. The introduction seems to be a little lacking in the emphasis on looking at socio-demographic factors in relation to alcohol consumption at the worldwide level. What are the common socio-demographic factors associated with alcohol consumption in college students worldwide - in different regions or different socio-demographic strata (for example, women)?

2. Similarly, the discussion is limited in the comparison of the sociodemographic determinants found here with global, and other regional data.

3. I was not sure why the data from “Isralowitz R and Hong O (1988) Singapore: a study of university students’ drinking behaviour. British Journal of Addiction 83(11), 1321–1323” not included in the present systematic review and meta-analysis. The paper seems to provide some usable data on ethnicity and gender.

---

## [Reviewer Report]

The manuscript addresses how ASEAN findings relate to global research. The authors note common risk factors found internationally and reference key global studies. However, I believe more explicit comparisons would strengthen the work.

Introduction

• I suggest expanding the background to detail international trends in alcohol consumption among university students. For example, when discussing factors like male gender and parental alcohol use (as seen on line 399), incorporating studies such as Karam et al. (2007) or Francis et al. (2014) would help set a clearer global context.

• I found that adding specific global examples here would position the ASEAN focus within a broader framework.

Methods

• I recommend clarifying whether global studies were considered during the initial search. On lines 125–140, if the authors mention that international databases or search terms were used, it would provide a comparative baseline.

• I suggest justifying the exclusion of non-ASEAN studies by referencing international research methods, which would help readers understand the rationale behind the regional focus.

Results

• I would like to see the meta-analysis findings compared with those from global studies. For instance, when presenting the odds ratios for gender (line 255), discussing how these figures align with international data could provide readers with a broader context.

• I feel that addressing heterogeneity by comparing regional and global data would further enrich the results section.

Discussion

• I found the discussion on the regional findings in relation to global research a bit underdeveloped. I suggest expanding this section to include a detailed comparison of intervention strategies used in ASEAN versus those implemented in North America or Europe.

• Addressing how cultural, policy, or economic differences might influence these results would make the global learning component more robust. I recommend discussing specific international studies to illustrate these points.

Conclusion

• I suggest summarizing how the regional findings contribute to global understanding. Emphasizing the potential for these results to inform both international policy and intervention strategies would highlight the broader impact of the research.

• Adding a statement on how future research could integrate regional specificity with global perspectives would be a valuable enhancement.

Enhancing these sections as suggested would better position the review within a global framework while maintaining its regional focus.

---

## [Editor Report]

Thank you for submitting your article to Global Mental Health. The reviewers recognize the importance of this systematic review and meta-analysis. However, they have identified some gaps, particularly in the introduction and discussion sections that could better situate this paper within the broader literature. They have also requested some points of clarification in the methods and results. We hope the authors will incorporate this feedback and consider revising and resubmitting this article.

---

## [Editor Report]

Thank you for your thorough revision of the manuscript, which have been responsive the comments from reviewers.